# Review on Preparation Technology and Properties of Refractory High Entropy Alloys

**DOI:** 10.3390/ma15082931

**Published:** 2022-04-17

**Authors:** Xiqiang Ren, Yungang Li, Yanfei Qi, Bo Wang

**Affiliations:** 1College of Metallurgy and Energy, North China University of Science and Technology, Tangshan 063210, China; rxq0219@126.com (X.R.); liyungang59322@163.com (Y.L.); 2Faculty of Materials and Manufacturing, Beijing University of Technology, Beijing 100124, China; wangbo@bjut.edu.cn

**Keywords:** refractory high entropy alloys, preparation technology, properties, radiation resistance

## Abstract

Refractory high entropy alloys have broad application prospects due to their excellent comprehensive properties in high temperature environments, and they have been widely implemented in many complex working conditions. According to the latest research reports, the preparation technology of bulk and coating refractory high entropy alloys are summarized, and the advantages and disadvantages of each preparation technology are analyzed. In addition, the properties of refractory high entropy alloys, such as mechanical properties, wear resistance, corrosion resistance, oxidation resistance, and radiation resistance are reviewed. The existing scientific problems of refractory high entropy alloys, at present, are put forward, which provide reference for the development and application of refractory high entropy alloys in the future, especially for plasma-facing materials in nuclear fusion reactors.

## 1. Introduction

High entropy alloys (HEAs) are a new type of metallic alloys, which are composed of at least five major metallic elements in equiatomic ratios or near-equiatomic ratios, and the atomic percentage content of each major element is 5~35% [1,2]. There are four core effects in HEAs: the high entropy effect in thermodynamics [3], the lattice distortion effect in structure [4], sluggish diffusion in kinetics [5,6], and the ‘cocktail’ effect in their properties [7,8,9,10]. The comprehensive properties of HEAs are superior to those of traditional alloys due to their four core effects. At present, HEAs are mainly divided into five categories: 3d transition HEAs, refractory HEAs (RHEAs), lightweight HEAs, precious HEAs, and rare-earth HEAs [3].

RHEAs, with excellent mechanical properties at elevated temperatures, are considered as promising high temperature application materials, which are suitable for complex working conditions such as atomic energy, aerospace, the military industry, and advanced nuclear reactors, etc. [11]. RHEAs were first put forward in 2010, which quickly attracted the attention of researchers due to their high strength at 1600 °C [12,13]. Initially, RHEAs are based on five refractory elements (Mo, Nb, Ta, V, and W), and then, five refractory elements, such as Ti, Zr, Hf, Cr, and Re, are added. Later, a small amount of non-refractory metal elements, such as Al, Si, Co, or Ni, are added to improve its comprehensive properties [14]. Usually, the main functions of specific elements in RHEAs are shown in Table 1, but the strengthening potential of one specific element depends on the others present. In some cases, one element might be an effective strengthener, while in other cases, it might not. Therefore, Table 1 can only be used as a reference for element selection in RHEAs, but the specific situation still needs specific analysis. The expansion of constitutive elements of RHEAs indicates that RHEAs have attracted more attention, and its application scope is more extensive. Thus, RHEAs have great significance and application prospects.

## 2. Preparation Technology of RHEAs

According to application fields and functions, the existing forms of RHEAs are mainly divided into bulk and coating/film. Due to the high melting points of constituent elements in RHEAs, the preparation technologies of RHEAs are limited compared with other HEAs. The main preparation technologies of bulk RHEAs are arc melting and powder metallurgy, and the major preparation technologies of coating/film RHEAs are laser cladding and magnetron sputtering, etc.

### 2.1. Preparation Technology of Bulk RHEAs

#### 2.1.1. Arc Melting

Arc melting is to melt metal through arc discharge generated at high temperature. Arc melting is a method of smelting metal or alloy, in a high purity argon atmosphere, by using electric energy to produce an arc between electrodes and the melted material (or electrodes). Firstly, the RHEAs components, with predetermined proportions, are put into a water-cooled copper crucible, according to the order of melting point, from low to high. Secondly, the furnace cavity is filled with protective argon gas. Finally, RHEAs are melted, and RHEAs with no residual raw material in the ingots and macroscopic chemical homogeneity are obtained by repeated melting.

Many RHEAs have been successfully fabricated by arc melting with high purity argon, such as HfNbTiZr [48], HfZrTiTa_0.53_ [49], and NbZrTiCrAl [50] RHEAs with single-phase BCC. The microstructure of dendrite and interdendritic NbZrTiCrAl RHEAs is inhomogeneous. Nb and Ti are mainly distributed in the dendrites, while Al, Cr, and Zr are mainly distributed in the interdendritic.

The advantage of arc melting is that it can effectively remove volatile impurities because of the high melting temperature. The method is suitable for preparing RHEAs with high melting points. However, due to the complex composition of RHEAs’ constituent elements with high melting points, there will be serious composition segregation in the smelting process [51]. Therefore, for chemical segregation, such as dendritic microstructure, it is mainly reduced by increasing the cooling rate of the solidification process, prolonged heat treatment at high temperature, or adjusting the chemical compositions of the ingots. 

#### 2.1.2. Powder Metallurgy 

Powder metallurgy (PM) is a method of preparing bulk RHEAs from raw materials (elemental powder or pre-alloyed powder) by ball milling/mixing, pressing, sintering, and subsequent treatment [52]. The raw material powder of powder metallurgy is prepared by mechanical alloying.

TiVNbTa RHEAs, sintered at 1100 °C, have excellent mechanical properties, and their compressive yield strength and plastic strain are 1506.3 MPa and 33.2%, respectively [53]. Two kinds of precipitates were detected in NbMoTaWVCr_0.6_ RHEAs. One is the tetragonal phase, rich in Ta, V, and O, precipitated at the grain boundaries and inside the grain, while the other is the irregular Laves phase (Cr,V)_2_(Ta,Nb), rich in Cr, Ta, Nb, and V, precipitated at the triple junctions. In addition, the crystal structure of the (Cr,V)_2_(Ta,Nb) Laves phase changes from C15 (≤1500 °C) to C14 (at 1600 °C). Figure 1 shows the TEM images and selected area diffraction pattern (SAED) of NbMoTaWVCr_0.6_ RHEAs, sintered at 1400 °C, 1500 °C, and 1600 °C, and the chemical compositions and constituent phase of NbMoTaWVCr_0.6_ RHEAs were show in Table 2. In addition, NbMoTaWVCr_0.6_ RHEAs, sintered at 1500 °C, exhibit excellent properties. Their compressive yield strength is 3416 MPa, plastic strain is 5.3%, and Vickers hardness is 9908 MPa [54]. 

Cao et al. [55,56] analyzed the characteristics of TiNbTaZrAl_x_ RHEAs prepared by arc melting and powder metallurgy. The results showed that the as-cast TiNbTaZrAl_x_ RHEAs, prepared by arc melting, are seriously segregated, and their microstructure is the BCC phase, rich in (Ta, Nb), and the BCC2 phase, rich in (Al,Zr). However, the composition of TiNbTaZrAl_x_ RHEAs prepared by powder metallurgy is uniform, and the microstructure is a single BCC phase.

The advantages of powder metallurgy are low cost, high efficiency, low preparation temperature, more uniform composition, and finer microstructure of RHEAs. The disadvantage is that the RHEAs prepared by this method are faced with the problems of pollution and internal oxidation, which reduce the properties of the RHEAs [51].

### 2.2. Preparation Technology of Coating/Film RHEAs

#### 2.2.1. Laser Cladding

Laser cladding is a new surface modification technology, which uses a high-energy laser beam to melt coating material and the surface layer of substrate, and then, a pore and crack-free coating is perfectly bonded to the substrate. There are many kinds of lasers, such as the YLS-4000-Y13 fiber laser system with 3000 W, JK802/1002 Nd: YAG laser (JK Lasers, Rugby, UK) with 1000 W maximum power, etc. This method can effectively improve the corrosion resistance, wear resistance, and oxidation resistance of substrate, which has been widely implemented in the field of coating manufacturing. The schematic diagram of laser cladding is shown in Figure 2.

AlTiVMoNb RHEAs’ coating, with single BCC phase structures were deposited on Ti-6Al-4V substrate by laser cladding [57]. The hardness of the coating is 885.5 HV_0.2_, which is 1.66 times that of the vacuum arc melting alloy and 2.52 times that of the substrate (353.1 HV_0.2_). Its oxidation weight is only 10.58% of that of the substrate, which means it has good oxidation resistance. Moreover, RHEAs’ coatings with excellent properties were prepared by laser cladding technology, and they are suitable for extreme working environments, such as MoFe_x_CrTiWAlNb_y_ [58], W_x_NbMoTa [59], and TiZrNbWMo [60], etc. 

Laser cladding technology can improve surface properties of substrate without changing the shape and inherent properties of substrate, and it can also repair the damaged substrate surface [61]. In addition, the superficial layer of the coating and the substrate is metallurgical bonding, and the bonding layer is compact. The advantages of laser cladding are simplicity, flexibility, time saving, and material saving, etc., which can effectively improve the surface properties of substrate [62]. Laser cladding technology can prepare RHEAs’ coatings, with excellent properties, on the substrate surface so as to improve the properties and market competitiveness of substrate.

#### 2.2.2. Magnetron Sputtering

Magnetron sputtering means that the target is bombarded by plasma in a vacuum environment, and the sputtered metal atoms are deposited on the substrate surface. The target provides atoms for the deposited metal film, so the purity of the film is related to the purity of the target. Therefore, the method requires a high purity target. The schematic diagram of magnetron sputtering is shown in Figure 3.

Li et al. [63] deposited a series of TaWTiVCr RHEA films by magnetron sputtering. The mechanical properties of Ta_24_W_24_Ti_16_V_19_Cr_17_ RHEA films were the best, with hardness of 27.54 GPa, modulus of 274.39 GPa, average friction coefficient of 0.34, and wear rate of 5.01 × 10^−9^ mm^3^/(N·mm). Therefore, it can be inferred that the RHEAs have excellent wear resistance. In addition, TaNbHfZr RHEA films with good plasticity [64], CuMoTaWV RHEAs films with high hardness [65], CrNbTiMoZr RHEA films with high wear resistance [66], and NbMoTaW RHEA films with good hardness and resistivity [67] were prepared by magnetron sputtering.

Our team prepared WTaTiCrV RHEAs films on the surface of silicon sheets by magnetron sputtering. The surface morphology and cross-section of WTaTiCrV RHEA films are shown in Figure 4. The surface of WTaTiCrV RHEA films is dense and smooth. The element distribution of the RHEA films is uniform, indicating that there is no element/phase segregation. However, WTaTiCrV RHEAs is non-equiatomic composition because the sputtering resistance of each atom is different.

RHEA films, composed of various elements, can be prepared by magnetron sputtering, and the films have high compactness. However, the utilization rate of the target is low, the film thickness is limited, and the adhesion between the film and the substrate is low. The anti-sputtering ability of each atom is different, so it is difficult to accurately control the component content of RHEAs. Therefore, solving these problems is of great significance for expanding the application of magnetron sputtering in the field of RHEAs film/coating.

## 3. Properties of RHEAs

### 3.1. Mechanical Properties

The excellent mechanical properties of metal materials are the basis of their application in various industries. Generally, at room temperature, the constituent elements of RHEAs with high strength include V, Nb, Ta, Cr, Mo, W, etc., and the constituent elements of RHEAs with good plasticity include Ti, Zr, Hf, etc. [68].

NbMoTaW RHEAs is a promising material for gas turbine engines due to its high strength, excellent thermal stability, and softening resistance at elevated temperatures. However, the brittleness of NbMoTaW RHEAs at room temperature limits its application and development. To solve the problem, Ti was added to NbMoTaW RHEAs, and Ti_x_NbMoTaW RHEAs was prepared by vacuum arc melting [69]. The test results show that the ductility and yield strength of Ti_x_NbMoTaW RHEAs were higher than that of NbMoTaW RHEAs (Table 3). In addition, as shown in Figure 5, the tensile test showed that the fracture changes from an intergranular fracture to a transgranular fracture [69]. The yield strengths of NbTaVW, NbTaTiV, and NbTaTiVW RHEAs were 1530 MPa, 965 MPa, and 1420 MPa, respectively, and their fracture plastic strain were 12%, >50%, and 20%, respectively [70]. Thus, it can be seen from literature [70] that W can increase its yield strength and decrease its ductility. On the contrary, Ti can improve its ductility and reduce its yield strength. The yield strength and comparable plasticity of single phase NbTaTiVZr RHEAs are higher than that of single phase NbTaTiV RHEAs [71]. Lee et al. [71] systematically and quantitatively studied that the increase in yield strength was caused by severe lattice distortion. The addition of Sc in RHEAs can not only reduce its density but also improve its high temperature strength. The literature reported that TiZrHfNbSc RHEA was a single-phase BCC, with a density of about 7.16 g/cm^3^, Vickers hardness of about 3800 MPa, yield strength of about 650 MPa, and compression deformation rate of >60% [72].

RHEAs should have excellent mechanical properties at high temperature and room temperature, and the density of RHEAs should be reduced as much as possible. The mechanical properties of RHEAs are closely related to its added elements. Some elements can improve the strength and plasticity of RHEAs, while some elements can only improve its strength or plasticity. Therefore, it is difficult to choose appropriate additive elements to balance the relationship between the strength and plasticity of RHEAs. To improve the properties of RHEAs by adding elements, we should fully consider the impact on its comprehensive properties and try to achieve balance between the properties.

### 3.2. Wear Resistance

Many conventional metal materials exhibit high strength, high ductility, corrosion resistance, wear resistance, etc., which are widely used in aerospace, automobile manufacturing, ship domain and other fields. Wear resistance is one of the basic properties of metal materials, which must be studied before being applied in the above fields. For extreme conditions, conventional metal materials cannot meet the requirements of wear resistance. Therefore, it is necessary to develop RHEAs with high wear resistance.

In the study of wear resistance of MoTaWNbV RHEAs and MoNbTaTiZr RHEAs, it is found that the wear resistance of two-phase (BCC + HCP) MoTaWNbV RHEAs is better than that of single-phase (BCC) MoNbTaVW RHEAs, and both of them are better than that of a commercial superalloy, Inconel 718, which is related to the crystal structure of RHEAs and the Ti and Zr elements in RHEAs that are easy to use to form lubricating oxides [73,74]. The CoCrNbNiW RHEA coating was deposited on 45 steels by laser cladding, and its wear loss and wear rate were 0.26 and 0.23 times of that of the substrate, respectively [75].The wear resistance of the RHEA coating is obviously better than that of the substrate. Therefore, the RHEA coating deposited on the substrate can significantly improve the wear resistance and market competitiveness of the substrate. 

Single-phase (BCC) TiNbZrMo HEA coating, with less wear loss and low wear rate, was prepared on 316 L stainless steel by laser cladding. Thus, the wear resistance of 316 L stainless steel was improved, and its service life in marine environment will be enhanced [76]. Guo et al. [77] found that NbTaWMo RHEAs are single-phase (BCC), and NbTaWMoSi_0.25_ RHEAs are two-phase, composed of BCC phase and silicide (Nb_5_Si_3_ and Ta_5_Si_3_). Within 25~800 °C, the improvement of wear resistance of NbTaWMoSi_0.25_ RHEAs is related to the silicide formed by Si doping. In addition, the wear mechanism of the RHEAs is affected by temperature. It is abrasive wear from low temperature to medium temperature, and it is abrasive wear and oxidation wear at high temperature.

Generally, materials with bad wear resistance are coated. Therefore, the RHEA coating can improve the wear resistance of substrate. RHEAs, which have high wear resistance from room temperature to high temperature, are more competitive in the market. The wear resistance of RHEAs’ coating is related to its constituent elements.

### 3.3. Corrosion Resistance

Corrosion is one of the key factors leading to metal material failure, which will cause equipment shutdown, economic losses, and personal safety hazards. With the rapid development of science and technology, the requirement of material properties in various fields is more and more stringent. It is imperative to develop materials with outstanding corrosion resistance. RHEAs are favored by researchers because of its excellent mechanical properties and wear resistance. If RHEAs have good corrosion resistance, it is of great significance to broaden their application field. Therefore, the research on high corrosion resistance of RHEAs has become one of current research hotspots.

Wen et al. [78] revealed that the corrosion resistance of NiCrCoTiV RHEAs is superior to 316 stainless steel in 3.5 wt.% NaCl solution, and the corrosion resistance of NiCrCoTiV RHEAs was improved after annealing at 500 °C for 18 h. In addition, in 1 mol/L H_2_SO_4_ solution, NiCrCoTiV RHEAs annealed at 700 °C had the best corrosion resistance, followed by 600 °C and 500 °C, which was due to the smallest grain size of RHEA precipitates after annealed at 700 °C, and the weaker galvanic effect between precipitates and eutectic structure, the better corrosion resistance of RHEAs. Yan et al. [79] studied the corrosion resistance of as-cast ReTaWNbMo RHEAs in 3.5 wt.% NaCl solution at different heat treatment temperatures (293 K, 673 K, 873 K, 1073 K, 1273 K) for 12 h. *I*_corr_ is proportional to the corrosion rate, so *I*_corr_ can directly display the corrosion resistance of materials in corrosive solutions. With the increase in annealing temperature, *I*_corr_ first increases and then decreases. The corrosion resistance of the RHEAs is the weakest at an annealing temperature 673 K, and the corrosion resistance of the RHEAs is the strongest at an annealing temperature 1273 K. The results showed that the corrosion resistance of ReTaWNbMo RHEAs is related to its heat treatment conditions. The corrosion resistance of VNbMoTaW RHEA coating is obviously improved by adding Cr and B elements [80]. In addition, the corrosion resistance of TiZrNbTaMo RHEAs is better than that of Ti6Al4V alloy and 316 L stainless steel, characterized by passivation and no pitting [81]. Relevant electrochemical parameters are shown in Table 4 and Figure 6.

The corrosion resistance of RHEAs is related to constituent elements, constituent ratio, and heat treatment conditions. The excellent corrosion resistance of RHEAs provides new ideas for the application of new materials as working parts in acid-base or seawater corrosive working environments, which is of great significance to improve the service life of equipment used in extreme working conditions.

### 3.4. Oxidation Resistance

RHEAs have good mechanical properties at high temperature environments, but their antioxidant capacity is weak, which limits their application and development. Nowadays, to improve the oxidation resistance of RHEAs, high temperature protective elements (Al, Si, Cr, Ti, etc.) are added to RHEAs to form protective layers (Al_2_O_3_, Cr_2_O_3_, SiO_2_, etc.). However, the high temperature protective elements and their addition amounts in RHEAs are closely related to the constituent elements of RHEAs. If the addition elements are not selected properly or the addition amount is not controlled properly, it will lead to the precipitation of unfavorable intermetallic compounds in RHEAs, such as Laves.

NbCrVWTa RHEAs showed excellent oxidation resistance at lower temperatures (600~800 °C). However, the oxidation resistance of NbCrVWTa RHEAs decreased, obviously, at higher temperatures (>1000 °C) [82]. The oxides of W and Ta are cylindrical, the oxides of Nb and Cr are granular or nodular (1200 °C), and the oxides of V appear to be whisker-like only at 1200 °C (Figure 7). To solve the poor oxidation resistance of TiZrNbHfTa RHEAs, 0~1 at.% Al was added to TiZrNbHfTa RHEAs [83], and the higher the Al content, the better oxidation resistance of Al_0-1_TiZrNbHfTa RHEAs. In Al_1_TiZrNbHfTa RHEAs, an effective barrier against oxidation was provided by comparatively stable oxide scales between 700 °C and 900 °C. However, the oxidation resistance becomes worse at 1100~1300 °C because the less dense oxide layer provides an effective diffusion channel for oxygen, which makes it easy for oxygen to penetrate the matrix. Müller et al. [84] investigated the oxidation resistance of TaMoCrTiAl(0,1)Si RHEAs. The results show that the oxidation resistance of TaMoCrTiAl1Si RHEAs was lower than that of TaMoCrTiAl RHEAs. That was because oxygen and nitrogen pass through the interface between Laves phase and BCC phase in the RHEAs during high temperature oxidation, and internal corrosion occurs along the phase boundary in the RHEAs. The oxidation resistance of TaMoCrTiAl, NbMoCrTiAl, NbMoCrAl, and TaMoCrAl RHEAs at 1000 °C in air were studied [85]. The results showed that the protective layers of Al_2_O_3_, Cr_2_O_3_, and CrTaO_4_ were formed in these RHEAs containing Ta. Ti promotes the formation of CrTaO_4_. In these RHEAs containing Nb, although protective layers of Al_2_O_3_ and Cr_2_O_3_ originally formed (Figure 8), Nb_2_O_5_ polytypes formed near the metal/oxide interface, which cause oxide scale and oxide spallation. 

The chemical composition of RHEAs is complex, and it is easy to form complex oxides in the oxidation process. These complex oxides will affect the oxidation resistance of RHEAs. The improvement of oxidation resistance of RHEAs is directly related to their application and development. Although the additions of Al, Cr, and Ti can improve the oxidation resistance of RHEAs by the formation of protective oxide layers, some refractory elements in RHEAs may destroy the oxide layers and reduce the oxidation resistance of RHEAs. Therefore, the selection of protective elements in RHEAs and the determination of their addition amount are one of the key issues to improve the oxidation resistance of RHEAs. In addition, the improvement methods and mechanism of oxidation resistance of RHEAs need to be further deepened.

### 3.5. Radiation Resistance

Nuclear fusion energy, with the advantages of safety, high efficiency, rich raw materials, and being clean, is considered as one of the ideal new energy in the future. Nuclear fusion energy brings opportunities for new energy and brings unprecedented new challenges to the structural materials of nuclear fusion devices. The structural materials of a nuclear fusion device should have excellent high temperature properties and radiation resistance to ensure its safe and stable operation in service. In addition, these structural materials should also have low activation characteristics so that the recovery, decomposition, and secondary development and application of nuclear reaction waste will further promote nuclear fusion energy to become a real clean energy.

The latest research showed that RHEAs exhibit better mechanical properties, thermodynamic properties, and physical properties than traditional alloys, and some RHEAs showed good radiation resistance and phase stability after ion irradiation. The irradiation experiment was carried out under 4.4 MeV high energy Ni^2+^ ions with a fluence of 1.08 × 10^17^ ions/cm^2^ at room temperature. The irradiation behavior of HfTaTiVZr RHEAs was studied and compared with 304 stainless steel [86]. It was found that the hardness and yield strength of HfTaTiVZr RHEAs increased by about 13% and 28%, while the hardness and yield strength of 304 stainless steel increased by about 50% and 54% (Figure 9). Therefore, HfTaTiVZr RHEAs have excellent irradiation resistance, which is attributed to the sluggish diffusion of atoms in RHEAs, which reduces the effective interstitial and vacancy mobility and limits the damage radiation. El-Atwani et al. [87] deposited W_38_Ta_36_Cr_15_V_11_ RHEAs films with single phase BCC crystalline structure by magnetron deposition. Transmission electron microscopy (TEM) and atom probe tomography (APT) analysis showed that there were Cr rich and V rich second phase particles in RHEAs films. At 1073 K, after 1 MeV (8 dpa) Kr^2+^ irradiation, no evidence dislocation loop induced by irradiation was found, and the irradiation hardening was small, so it had excellent radiation resistance. In addition, the RHEAs were suitable for batch production, so they can be ideal structural materials for extreme irradiation conditions. After irradiation with 300 keV Ni^+^ at 373 K, HfNbTaTiZr RHEAs produced radiation swelling. There was no phase transformation and almost no change in nano hardness before and after irradiation. Thus, HfNbTaTiZr RHEAs have good resistance to high dose irradiation [88]. Moschetti et al. [89] studied the irradiation behavior of near-equiatomic TiZrNbHfTa RHEAs at room temperature, with a He^2+^ ion at an energy of 5 MeV and a flux of 1.6 × 10^12^ ion/(cm^2^·s) to a fluence of 4.4 × 10^17^ ion/cm^2^. The results showed that the hardness of recrystallized and nanocrystalline RHEAs increase slightly, the yield strength and ultimate tensile strength increase significantly, but the ductility does not lose, which is related to irradiation hardening and the transformation from shear localized failure of smooth fracture surface to fracture morphology composed of fine dimples and intergranular failure features.

For RHEAs working in nuclear fusion reactor, the property requirements are more stringent. The ion irradiation resistance of RHEAs is better than that of traditional alloys, but the irradiation mechanism of RHEAs needs further study. In addition, the research on its thermal irradiation resistance and neutron irradiation performance should be supplemented. This is of great theoretical significance to further improve the application of RHEAs in the field of nuclear fusion. The alloy elements of plasma-facing materials in a nuclear fusion reactor will produce radioactive nuclides under high-energy neutron irradiation, and the half-life of different elements is different. The research on RHEAs in nuclear energy mainly focuses on RHEAs composed of elements with long half-lives after neutron irradiation, such as Mo, Zr, Nb, and Co. However, these elements are not conducive to the subsequent treatment of nuclear reaction waste and the repair of a furnace wall. There are few studies on RHEAs composed of elements with short half-life, and the RHEAs composed of metal elements with relatively short half-life are required for nuclear fusion devices, which is conducive to the recovery, decomposition, and secondary development and application of nuclear reaction waste. Only a few elements are suitable for the preparation of low activity materials, such as Fe, Cr, V, Ti, W, Ta, and C. Moreover, He, the product of nuclear fusion reaction, has low reactivity. Long-term retention in materials will lead to material swelling or embrittlement, which will cause great hidden dangers to the normal operation of a nuclear fusion reactor. After one year of fusion reactor operation, the He concentration of various metal elements in different parts of the fusion device was detected. It was found that the helium production concentration of Fe, Cr, V, Ti, W, and Ta was low [90]. Besides, W and Ta have high resistance to radiation-induced embrittlement and swelling. Through rapid and significant mutual diffusion, Ti plays an important role in improving the sintering density, reducing RHEAs density, and improving the oxidation resistance and corrosion resistance of RHEAs [91]. V can improve the strength and hardness of RHEAs and promote the formation of Laves phase intermetallic compounds [92,93]. Cr can improve the oxidation resistance of RHEAs and reduce its density. Cr has strong interactions with other refractory elements, that is, large negative mixing enthalpy, which can improve the enthalpy of RHEAs. The precipitation of the Cr-rich Laves phase on the BCC solid solution matrix can further improve the high temperature strength of the alloy [94]. 

RHEAs are one of the promising candidate materials for nuclear fusion devices because they can meet the requirements of radiation resistance in complex environments. The radiation resistance of RHEAs is better than that of traditional alloys, which is closely related to the sluggish diffusion of atoms effect and lattice distortion effect. The RHEAs composed of W, Ta, Ti, V, and Cr are candidate materials for nuclear fusion reactor. There are few reports on the thermal irradiation resistance and mechanism of RHEAs, so it is necessary to strengthen the research in this field to enrich the research theory of RHEAs as nuclear fusion structural materials and lay a theoretical foundation for its application in nuclear fusion structural materials.

## 4. Conclusions and Outlook

RHEAs, with excellent comprehensive properties, have broad application prospects in the fields of aviation, aerospace, and nuclear reactors with complex working conditions. According to preparation technology and service environments, RHEAs are mainly divided into block RHEAs and coating/film RHEAs. For bulk RHEAs, arc melting is suitable for producing an RHEA with higher temperatures, but its ingot is prone to serious composition segregation. Therefore, solving the problem of composition segregation is the basis and key to promote this method. Although repeated melting and reasonable heat treatment can effectively improve the macroscopic chemical homogeneity of RHEAs, it increases the workload. It is considered to add an electromagnetic stirring device to the arc melting furnace to improve the mixing of the solution by controlling the magnetic field. The chemical composition of RHEAs prepared by powder metallurgy is uniform and the microstructure is fine, but the pollution and internal oxidation reduce the comprehensive properties of RHEAs. The bottleneck of powder metallurgy is to improve the purity of RHEAs and reduce the internal oxidation, so it can be considered to add oxidation resistant elements to RHEAs to solve the problem.

It is reported that RHEA coating/film has excellent mechanical and physical properties, which can improve the surface properties of the substrate, repair the damaged surface of the substrate, and improve the comprehensive properties and market competitiveness of the substrate. The preparation technology of RHEA coatings/films are mainly focuses on laser cladding and magnetron sputtering. However, these two technologies cannot meet all the preparation and working environments of RHEA coatings/films. Therefore, it is necessary to introduce other coating/film preparation technologies in RHEA coatings/films to improve the application field and market competitiveness of RHEAs.

How to improve the basic properties of RHEAs, such as mechanical properties, wear resistance, corrosion resistance, and oxidation resistance, had been reported. RHEAs have excellent mechanical properties at high temperatures. It is necessary to pursue excellent mechanical properties at room temperature and reduce its density as much as possible. The constituent elements and phase composition of RHEAs will affect its microstructure, wear resistance, corrosion resistance, and oxidation resistance. According to the property requirements of RHEAs, the microstructure of RHEAs can be improved by adding appropriate elements, their composition proportion, and being equipped with reasonable heat treatment process. In addition, the effect of added elements on other properties of RHEAs should be considered to improve the comprehensive properties of RHEAs. 

RHEAs are one of the promising candidate materials for nuclear fusion reactor. The high temperature resistance, low hydrogen isotope retention, helium irradiation resistance, and low activation neutron irradiation of W, Ta, Ti, V, and Cr are conducive to their priority as candidate metal elements in low activation RHEAs in future nuclear fusion reactors. Therefore, the RHEAs composed of the above five elements are expected to become one of the candidate materials for plasma-facing materials, which will further promote nuclear fusion energy to become a real green energy.

## Figures and Tables

**Figure 1 materials-15-02931-f001:**
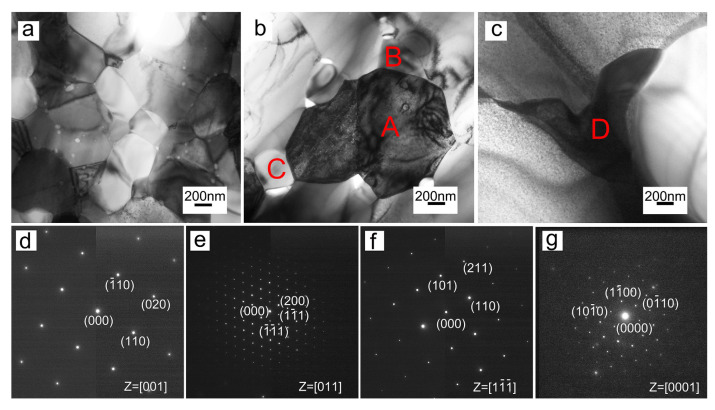
TEM images of bulk NbMoTaWVCr_0.6_ RHEAs sintered at different temperatures: (**a**) 1400 °C, (**b**) 1500 °C, (**c**) 1600 °C, (**d**) SAED pattern of grain A in (**b**), corresponding to a BCC structure, (**e**) SAED pattern of grain B in (**b**), corresponding to a C15 Laves phase (cubic), (**f**) SAED pattern of grain C in (**b**), corresponding to Ta_2_VO_6_ (tetragonal phase), (**g**) SAED pattern of grain D in (**c**), corresponding to a C14 Laves phase (hexagonal) [54]. (Reprinted with permission from Ref. [54]. Copyright 2018 Elsevier).

**Figure 2 materials-15-02931-f002:**
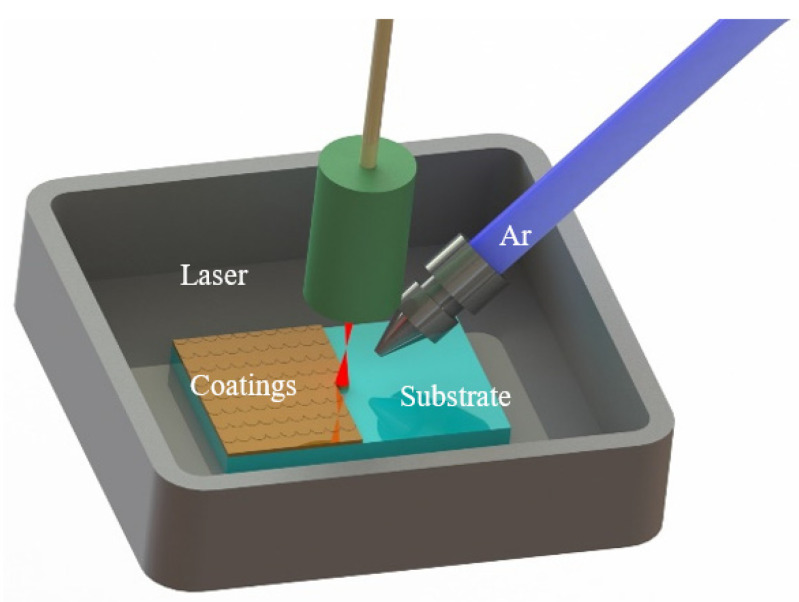
Schematic diagram of laser cladding.

**Figure 3 materials-15-02931-f003:**
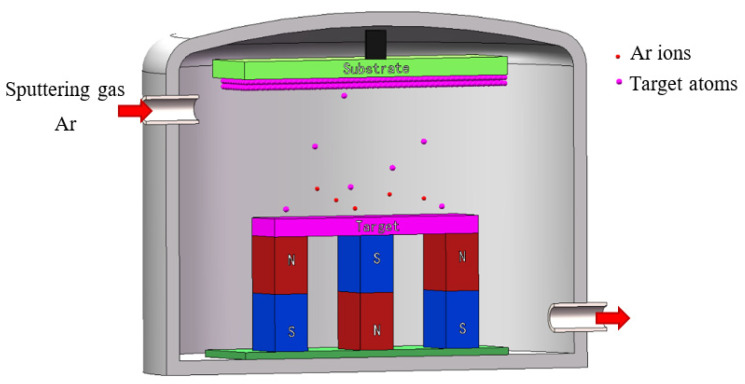
Schematic diagram of magnetron sputtering.

**Figure 4 materials-15-02931-f004:**
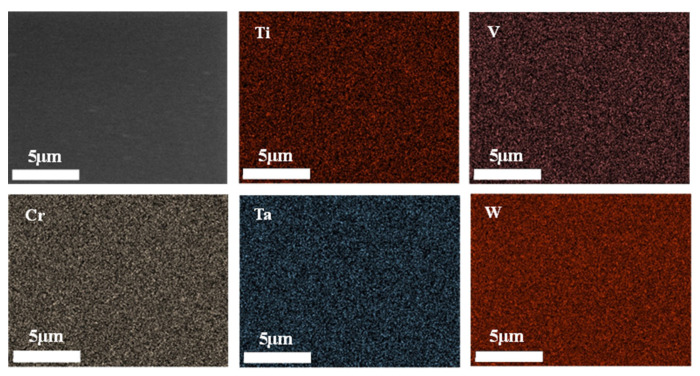
Characterization of WTaTiCrV RHEAs films on Si substrate with SEM-surface morphology and EDS elemental mapping of surface.

**Figure 5 materials-15-02931-f005:**
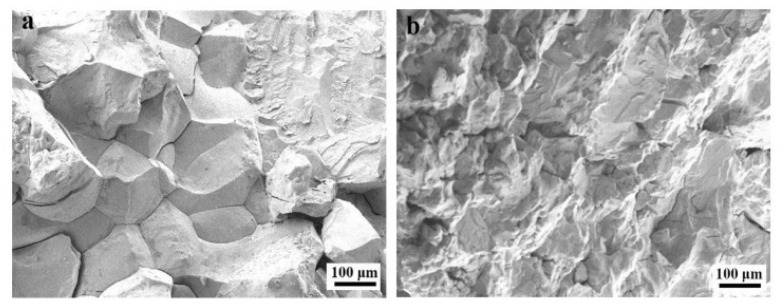
The SEM images of fracture surfaces of NbMoTaW RHEAs and Ti_x_NbMoTaW RHEAs [69]: (**a**) intergranular fracture; (**b**) transgranular fracture. (Reprinted with permission from Ref. [69]. Copyright 2017 Elsevier).

**Figure 6 materials-15-02931-f006:**
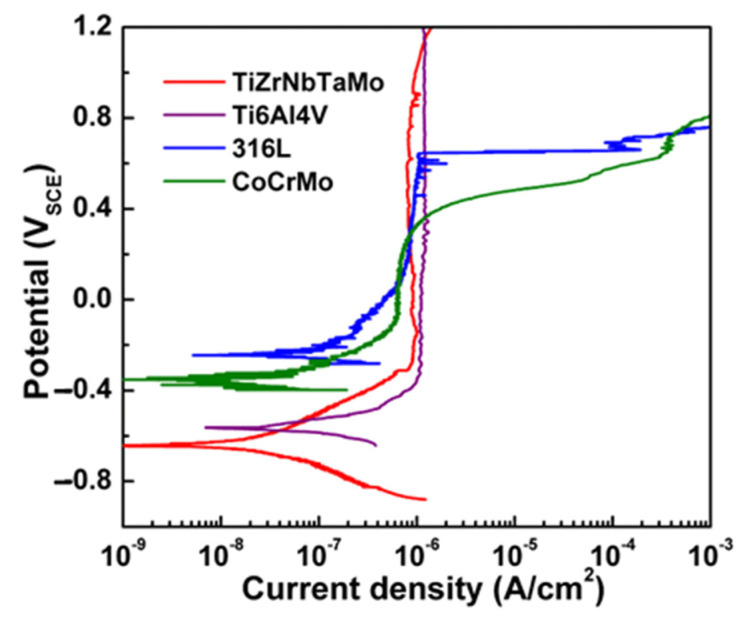
Potentiodynamic polarization curves of materials in phosphate buffer solution at 37 °C [81]. (Reprinted with permission from Ref. [81]. Copyright 2016 Elsevier).

**Figure 7 materials-15-02931-f007:**
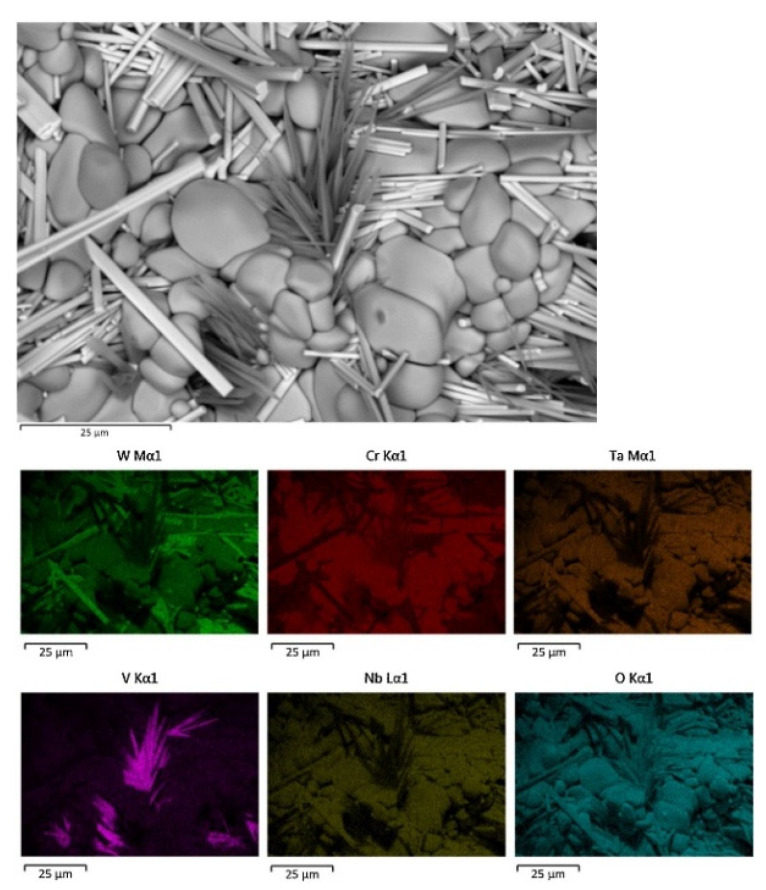
Energy-dispersive X-ray mapping for the microstructures of the oxides, developed at 1200 °C for 24 h [82]. (Reprinted with permission from Ref. [82]. Copyright 2020 Elsevier).

**Figure 8 materials-15-02931-f008:**
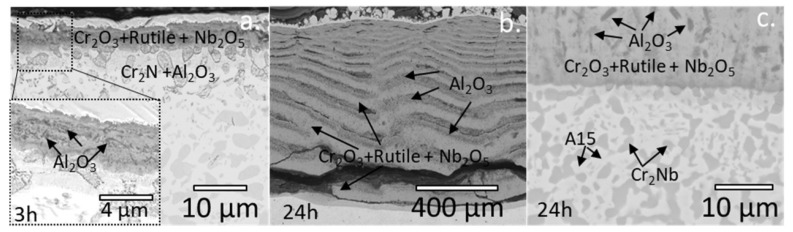
BSE images of NbMoCrAl after 3 h (**a**) and 24 h (**b**) of exposure to air at 1000 °C. A higher magnification BSE image of the metal/oxide interface of (**b**) is displayed in (**c**) [85]. (Reprinted with permission from Ref. [85]. Copyright 2019 Elsevier).

**Figure 9 materials-15-02931-f009:**
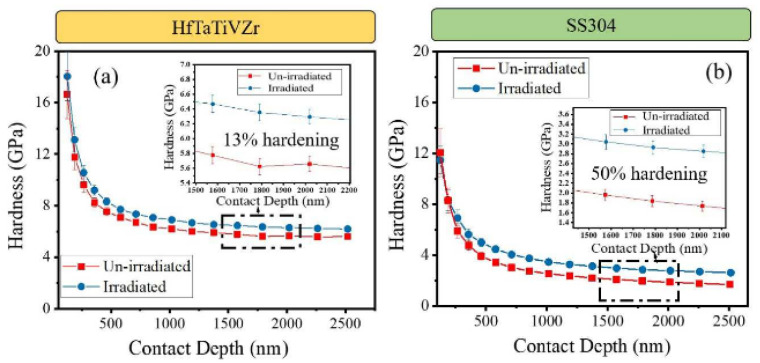
Hardness versus depth for irradiated and un-irradiated samples of (**a**) HfTaTiVZr high entropy alloy and (**b**) 304 stainless steel (SS304) [86]. (Reprinted with permission from Ref. [86]. Copyright 2019 Elsevier).

**Table 1 materials-15-02931-t001:** Elemental characteristics and function of RHEAs components.

Element Composition of RHEAs		Melting Point (°C)	Density at RT (g/cm^3^)	Main Performance
Refractory metal elements	Mo	2610	10.22	Hardness [15], Strength [16,17]
	Nb	2468	8.57	Hardness [18,19], Yield strength [20]
	Ta	2996	16.65	Strength [21]
	V	1902	6.11	Strength [22], Ductility [23], Hardness [24]
	W	3410	19.35	Hardness [25], Yield strength [26]
	Ti	1660	4.51	Oxidation resistance [27], Hardness [28], Yield strength [29], Plasticity [30], Ductility [21]
	Zr	1852	6.51	Oxidation resistance [31], Yield strength [32], Plasticity [33]
	Hf	2227	13.31	Hardness, Yield strength [34]
	Cr	1857	7.19	Oxidation resistance [35], Hardness [36], Corrosion resistance [37], Plasticity, Strength [38]
	Re	3180	21.04	Plasticity, Creep resistance [39]
Non-refractory metal elements	Al	660	2.7	Oxidation resistance [40], Strength, reduce density [41], Wear resistance, Yield strength [42]
	Si	1414	2.33	Oxidation resistance [43], Hardness, Strength [44]
	Co	1495	8.9	Yield strength [45], Wear resistance [46]
	Ni	1453	8.9	corrosion resistance [47], Yield strength [45]

Note: RT-room temperature.

**Table 2 materials-15-02931-t002:** The chemical compositions (in at.%) and constituent phases (%) of bulk NbMoTaWVCr_0.6_ RHEAs sintered at 1500 °C and 1600 °C [54]. (Reprinted with permission from Ref. [54]. Copyright 2018 Elsevier).

Nominal Composition		1500 °C			1600 °C		
		BCC PHASE	t-(Ta,V)O_2_	C15 Laves	BCC Phase	t-(Ta,V)O_2_	C14 Laves
Nb	16.67	16.04	2.45	17.83	16.37	2.16	16.65
Mo	16.67	17.28	-	3.59	16.70	-	3.01
Ta	16.67	15.38	24	18.92	16.18	22.43	22.31
W	16.67	17.91	-	7.47	17.68	-	-
V	16.67	16.17	12.96	11.58	16.02	13.41	13.17
Cr	16.67	17.21	-	40.58	17.04	-	44.82
O	-	-	60.57	-	-	61.97	-

**Table 3 materials-15-02931-t003:** Room-temperature yield strength (σ_0.2_), peak strength (σ_p_), and plastic strain (ε_p_) of the as-cast Ti_x_NbMoTaW RHEAs [69]. (Reprinted with permission from Ref. [69]. Copyright 2017 Elsevier).

RHEAs		σ_0.2_ (MPa)	σ_p_ (MPa)	ε_p_ (%)
Ti_x_NbMoTaW	Ti_0_	996	1148	1.9
	Ti_0.25_	1109	1197	2.5
	Ti_0.5_	1211	1578	5.9
	Ti_0.75_	1304	1593	8.4
	Ti_1_	1455	1910	11.5

Note: the molar ratio x = 0, 0.25, 0.5, 0.75 and 1.

**Table 4 materials-15-02931-t004:** Electrochemical properties of materials, in phosphate buffer solution, at 37 °C [81]. (Reprinted with permission from Ref. [81]. Copyright 2016 Elsevier).

Materials	TiZrNbTaMo	Ti6Al4V	316 L SS	CoCrMo
*E*_corr_ (mV_SCE_)	−607 ± 55	−571 ± 11	−234 ± 13	−320 ± 30
*I*_p_ (μA/cm^2^)	0.89 ± 0.06	0.96 ± 0.21	0.83 ± 0.03	0.42 ± 0.19
*E*_pit_ (mV_SCE_)	-	-	675 ± 30	435 ± 23

Note: *E*_corr_ = corrosion potential, *I*_p_ = passive current density, *E*_pit_ = pitting potential.

## Data Availability

Not applicable.

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
