# Peer review of "Review on Preparation Technology and Properties of Refractory High Entropy Alloys"

_materials, 2022, doi:10.3390/ma15082931_

Round 1

Reviewer 1 Report

Please see the enclosed file.

Author Response

Dear Reviewer 1,

We have studied the valuable comments from you carefully, and tried our best to revise the manuscript. The point to point responds to the reviewer’s comments are listed as following: 

Reviewer 2 Report

The manuscript’s subject is generally interesting due to its novelty, and the text is well organized into paragraphs. The review form and wide analysis of the available literature make it useful in the further development of high entropy alloys and their applications. The introduction provides a good, generalized background of the topic indicating the most important properties of refractory high entropy alloys that are widely discussed in the following sections of the manuscript. However, some explanations and minor corrections are needed to improve the quality of the manuscript.

  • In the description of Figure 1 or the text, there is no reference/discussion of the electron diffraction patterns placed next to SEM images. Please add a short explanation.
  • The accuracy of the results in Table 2 seems to be too high concerning the deviation indicated for some numbers. Please analyze it. Is it your results, or is the table reprinted from another paper? If yes, please add the reference.
  • Is Figure 3 made by the Authors of the Manuscript or copied? If so, please add a source or reference. This suggestion relates to tables and figures in the whole manuscript, where the source has not been indicated.
  • Page 4, last paragraph: “… single-phase BCC lattice…” - do the Authors mean the single-crystalline phase with the BCC structure? Please explain.
  • Figure 7: Please standardize the scale bar for all presented maps.
  • Please adapt the text to the journal’s requirements, e.g., figure instead of fig. or form of the reference numbers (without a superscript).
  • Slight language corrections are also needed (e.g., page 3, line 3: “segregation in the during”; page 12, line 5: “RHEAs has better”.

Author Response

Dear Reviewer 2,

We have studied the valuable comments from you carefully, and tried our best to revise the manuscript. The point to point responds to the reviewer’s comments are listed as following: 
